# Chemical Diversity of Flavan-3-Ols in Grape Seeds: Modulating Factors and Quality Requirements

**DOI:** 10.3390/plants11060809

**Published:** 2022-03-18

**Authors:** Guillermo F. Padilla-González, Esther Grosskopf, Nicholas J. Sadgrove, Monique S. J. Simmonds

**Affiliations:** Royal Botanic Gardens, Kew, Richmond Surrey, London TW9 3AB, UK; f.padilla@kew.org (G.F.P.-G.); e.grosskopf@kew.org (E.G.); n.sadgrove@kew.org (N.J.S.)

**Keywords:** procyanidin, *Vitis vinifera*, flavanol, antioxidant, abiotic, cultivar

## Abstract

Grape seeds are a rich source of flavan-3-ol monomers, oligomers, and polymers. The diverse profile of compounds includes mainly B-type procyanidins (especially C4→C8 linked molecules) and the key monomers, catechin, and epicatechin that are positively implicated in the ‘French Paradox’. Today grape seed nutraceuticals have become a multi-million-dollar industry. This has created incentives to elucidate the variations in chemistry across cultivars, to identify signs of adulteration, and to understand the intrinsic and extrinsic factors controlling the expression of metabolites in the seeds’ metabolome. This review provides a critical overview of the existing literature on grape seed chemistry. Although the biosynthetic pathways for polymeric procyanidins in seeds have not yet been explained, abiotic factors have been shown to modulate associated genes. Research of extrinsic factors has demonstrated that the control of procyanidin expression is strongly influenced, in order of importance, by genotype (species first, then variety) and environment, as claimed anecdotally. Unfortunately, research outcomes on the effects of abiotic factors have low certainty, because effects can be specific to genotype or variety, and there is limited control over physical metrics in the field. Thus, to gain a fuller understanding of the effects of abiotic factors and biosynthetic pathways, and realise potential for optimisation, a more fundamental research approach is needed. Nevertheless, the current synthesis offers insight into the selection of species or varieties according to the profile of polyphenols, as well as for optimisation of horticultural practices, with a view to produce products that contain the compounds that support health claims.

## 1. Introduction

The grapevine (*Vitis vinifera* L.) is one of the oldest and most significant crops in human civilisation [1]. The earliest evidence of human use comes from 7400–7000 year-old artefacts, excavated from remnants of the ancient Neolithic civilisations of north-western Iran, now known by archaeologists as Hajji Firuz Tepe (in the Zagros mountains) [2,3]. However, it was only two thousand years ago that the grapevine was introduced to the rest of the world. Today approximately 78 million metric tons of grapes are processed annually, with more than half going toward wine production and the remainder sold as fruit and culinary or nutraceutical items, such as grape seed extracts [4].

Until recently, grape seed was regarded as merely a by-product of the juice and wine industry. Nevertheless, they are the main source of flavan-3-ols in wine [5]. These flavan-3-ols are composed of a diversity of monomeric catechins and oligomeric or polymeric procyanidins that play an important role in the quality and characteristics of wine. Aside from their function as natural preservatives, they influence the aesthetics of wine, by determining its astringency, bitterness, ‘structure’, and maturation [6]. Furthermore, grape seed flavan-3-ols confer diverse biological effects in people and contribute to the healthful properties of wine. Indeed, since the early 1980s, there has been increasing evidence of an association between grape procyanidins and lower incidences of coronary heart disease (CHD) and atherosclerosis [7], contributing to the phenomenon termed “the French Paradox”. Accordingly, a high consumption of dietary fats among the French does not correlate to coronary heart disease (CHD) as expected, and it is believed that this is because of their intake of wine. Substantial evidence in favour of this paradox conveys a role for oligomeric procyanidins (up to pentamers) [7]. A seminal study demonstrated a possible mechanism involving inhibition of endothelin-1 (ET-1) synthesis, a vasoactive peptide that is associated with the development of coronary atherosclerosis [8,9]. Corder et al. [8] added that wines from Nuoro and the Gers area in France have up to four-fold greater ET-1 inhibitory activity and a higher oligomeric procyanidin content than wines from other regions [8]. Because the local people have lower incidences of CHD and an older median age, this strengthens the observation that procyanidins have a positive effect in the context of cardiovascular health.

In addition to cardiovascular protection, grape seed procyanidins have demonstrated various other in vitro and in vivo outcomes that suggest symptomatic improvement in cases of diabetes, obesity, cancers, and inflammation, among others [6,10,11,12]. However, although the biological properties of grape seed polyphenols have been extensively reviewed in recent years, information on the chemical diversity of grape seed flavan-3-ols and the genetic and climatic factors modulating their expression in different grape cultivars is sparce. This information void is problematic because variation in the types of procyanidins present in the seeds can elicit different biological effects.

Due to their high antioxidant potential and other biological properties, grape seed extracts are commonly used as ingredients in dietary supplements [13,14,15]. Previous studies reported that grape seed procyanidins possess higher antioxidant activities than other well-known antioxidants such as vitamin C [16,17]. However, as the popularity of grape seed products increase, there are concerns about the safety, quality, and efficacy of these products, due to lack of phenetic data on differences of chemical composition across species and varieties [18]. Even though procyanidins (any degree of polymerisation) are often listed as the active ingredient in grape seed dietary supplements, it is ultimately the oligomeric fraction (two to five degrees of polymerisation) that is the most important in the health context [19]. Experimental evidence suggests that dimeric to pentameric procyanidins are absorbed into the bloodstream, while molecules with higher degrees of polymerisation pass through the digestive system mostly unabsorbed [20,21,22]. Therefore, the procyanidin content *per se* is not a true reflection of efficacy.

Despite the promising health benefits associated with the polyphenolic fraction of grape seeds, studies rarely characterise the exact active compounds in these mixtures or measure their relative potencies. Furthermore, the profile of metabolites that are accumulated in grape seeds is different to those accumulated in the pulp and skin. These data are important if comparisons are to be made among grape seed extracts. The current review provides a critical and comprehensive synthesis of the chemical diversity of grape seed flavan-3-ols and makes recommendations to improve the quality of future chemical studies. An explanation of the importance of procyanidin structure in shaping bioactive outcomes is provided, to create clear parameters on extract ‘quality’. The main enzymes and intermediates involved in the biosynthesis of procyanidins are summarised and knowledge gaps are identified. Furthermore, the current review also consolidates the contribution of intrinsic (genetic) or extrinsic (environmental) factors on the chemistry of grape seeds and discusses how these sources of variation can be used to deliberately modulate the chemistry of grape seed extracts to optimise for quality. Finally, the quality of commercial grape seed extracts is variable, and the problems of adulteration mean that greater scrutiny is required as part of the quality control of grape seed products. Therefore, we also suggest detailed guidelines to ensure the quality, authenticity, and efficacy of commercial products.

## 2. Grape Seed Chemistry

Polyphenolic compounds represent the most abundant and chemically diverse group of metabolites accumulated in grape seeds and the third most abundant constituents in the whole fruit, after carbohydrates and organic acids [16]. Among the polyphenols, the diverse group of flavan-3-ols, such as monomeric catechins and procyanidins (also known as condensed tannins), represent the main chemical subclasses accumulated in grape seeds. Previous studies suggest that grape seeds contain between 60 to 70% extractable phenolic compounds, while the pulp and skin contain 10 and 28–35%, respectively [16]. Other chemical classes such as phenolic acids, stilbenes, flavonols, hydrolysable tannins (including gallotannins and ellagitannins), and organic acids have also been reported in grape seeds, usually as minor components that differ according to variety or species (see Section 2.2).

In the present section, the chemical diversity of grape seeds is summarised, giving details from the first reports made in the late 1960s through to early 2020. This information was retrieved from the SciFinder and Google Scholar databases using the key words: “grape seed”, “chemistry”, “flavanol”, and “procyanidins”. Additionally, the main enzymes and intermediates involved in the biosynthesis of procyanidins are discussed and knowledge gaps are highlighted. Grape seed quality is strongly related to biological effects and the specific end-use or intended application, so a discussion of pharmacokinetics and structure–activity relationships is also provided.

Most of the quality characteristics of wine, as well as the biological and nutraceutical properties are attributed to the polar components in grape seed extracts, particularly the flavan-3-ols, so the current review focuses on these components. Detailed reviews of the chemical constituents in grape seed oil are provided by [23,24,25].

### 2.1. Flavan-3-Ols

Flavan-3-ols represent the most reduced form of the flavonoids. They are characterised by the presence of a 2-phenyl-3,4-dihydro-2H-chromene skeleton, which is hydroxylated at position 3 of ring C (Figure 1). Favan-3-ols have two chiral centres at positions 2 and 3, which creates chiral diastereomers, giving four possible configurations: two enantiomers for each epimer. The enantiomers in the *trans* configuration are known as (+)- or (−)-catechin, and in the *cis* configuration they are (+)- and (−)-epicatechin (Figure 1). However, due to biosynthetic constraints, only one enantiomer for each epimer has been reported, which are (+)-catechin (2*R*, 3*S*) and (−)-epicatechin (2*R*, 3*R*) [26,27,28,29]. Incidentally, these diastereomers are the most common flavan-3-ol isomers reported in nature. Their opposite enantiomers (−)-catechin (2*S*, 3*R*) and (+)-epicatechin (2*S*, 3*S*) (Figure 1) (also known as *ent-*catechin or *ent*-epicatechin) are significantly less common [13]. However, the optical rotation and absolute configuration of monomeric grape seed flavan-3-ols cannot be deduced in routine mass spectral or NMR analysis, so they are not commonly specified in published studies.

The addition of a third hydroxyl group on ring B of the epimers (+)-catechin and (−)-epicatechin produces the monomers (+)-gallocatechin and (−)-epigallocatechin, respectively (Figure 1). These metabolites are commonly reported in the skins of the grape berry [6,30]. Furthermore, the esterification of gallic acid makes a ‘gallate’, which is a process known as galloylation.

Thus far, the chemical structures of eleven monomeric flavan-3-ols (Figure 1) have been reported in grape seed [28,31,32,33]. The monomeric flavan-3-ols (+)-catechin, (−)-epicatechin and (−)-epicatechin 3-*O*-gallate, represent the three most abundant and commonly reported metabolites [30,34]. The other less abundant monomers include (+)-catechin-3-*O*-gallate, (+)-gallocatechin, (−)-epigallocatechin, (+)-gallocatechin-3-*O*-gallate, (−)-epigallocatechin-3-*O*-gallate, and (−)-epigallocatechin-3-*O*-vanillate [19,31,35,36]. Flavan-3-ols are also found as sugar-linked molecules, but only as minor components in grape seed extracts. The two main glycosylated flavan-3-ols are (+)-catechin-4′-*O*-β-glucoside and (+)-catechin-7-*O*-β-glucoside [33] (Figure 1).

Out of the less common monomers, there are some reports of (+)-gallocatechin, (−)-epigallocatechin and their *O*-gallate derivatives in grape seeds [31,36], but the legitimacy of these findings is in doubt, because these metabolites are expressed in high concentrations in the skins of the grape berry. Several authors claim that these compounds are not present in the seeds, and that traces are from the fruit pulp or skin that has not adequately been removed prior to analysis [6,29,30,37]. However, it is also possible that variations in the chemistry among cultivars explain these differences. For example, some authors have detected (+)-gallocatechin, (−)-epigallocatechin, and their *O*-gallate derivatives in the seeds of some grape varieties by chromatographic comparisons with pure standards [19,31,36]. Significant differences were also observed in the content of epigallocatechin in the seeds of 70 grape varieties, ranging from 219 ± 44 to 1813 ± 124 μg·g^−1^ [36]. These galloylated metabolites are known to influence the flavour of wines and elicit specific biological effects [38,39,40]. Thus, understanding whether the abundance of these compounds can be regulated by intrinsic (genetic) or extrinsic (environmental) factors is an important aspect (discussed in Section 3).

It is only in the last decade that glycosylated catechins have been assigned in grape seeds [41]. According to Delcambre and Saucier [41], there are 14 types of flavan-3-ol monoglycosides, which are made from four aglycone units: (+)-catechin, (−)-epicatechin, (−)-epigallocatechin, and (−)-epicatechin gallate, but their structures were not resolved. These structures were partially identified by detailed interpretation of fragmentation patterns in targeted MS/MS experiments. However, in that first report, the position of the hexoside substituent was not determined. Furthermore, it was unknown if these compounds can also occur as constitutive units of oligomeric procyanidins.

A recent study by Zerbib et al. [33] confirmed the presence of flavan-3-ol monoglycosides in the seeds of three grape varieties: ‘merlot’, ‘cabernet-sauvignon’, and ‘syrah’. They identified five isomers of monomeric hexosides and four isomers of dimeric hexosides using LC-MS/MS analysis of seed extracts. Furthermore, standards of (+)-catechin-4′-*O*-β-glucoside and (+)-catechin-7-*O*-β-glucoside were obtained through enzymatic hemisynthesis and their structures confirmed by uni- and bi-dimensional NMR spectroscopy [33]. Hence, the presence of these two compounds in grape seeds, skins, and wines was confirmed using authenticated standards in LC-MS/MS.

Zerbib et al. [33] also identified glycosylated procyanidins in grape seeds, however, the glycosylation position and the stereochemistry of these molecules has not yet been resolved. A more recent study by Pérez-Navarro et al. [42] expanded the reports of glycosylated flavanols to 20 monomeric flavanol monoglycosides, four diglycosylated monomers, and three dimeric flavanol monoglycosides in the seeds and skins of six grape varieties [42]. However, the 2D structures of these glycosylated molecules need to be resolved spectroscopically. In another recent study the presence of glycosylated flavan-3-ols were also reported in five red wines from the ‘tannat’, ‘Alicante’, ‘merlot’, ‘syrah’, and ‘grenache’ grape varieties [33]. However, most of the glycosylated flavan-3-ols that have been reported in grape have not been comprehensively elucidated. In moving forward, the occurrence of monomeric and dimeric flavanol hexosides in wines should also be considered in the context of flavour and nutrition.

#### 2.1.1. Procyanidins

Grape seed flavan-3-ols are found as monomers, dimers, oligomers (3 to 10 units), and polymers (>10 units: i.e., condensed tannins) [5]. The condensation of (+)-catechin and/or (−)-epicatechin units produces procyanidins, and gallocatechins produce prodelphinidins. The literature indicates that grape seeds only express procyanidins, whereas prodelphinidins are restricted to grape skins [43]. The absence of prodelphinidins in grape seed raises additional doubts as to whether the reports of their constitutive monomer units (gallocatechins) in the seeds are accurate. There are 17 known procyanidin dimers, 16 trimers, and 1 tetramer. Together with the monomeric flavan-3-ols, this gives a total of 45 flavan-3-ols that have been reported in grape seeds (several other tentative structures await spectroscopic confirmation).

Based on the interflavanic linkage of procyanidins, they are often classified as B-type, if a single linkage occurs in C4→C8 or C4→C6 (Figure 2), or A-type, if, in addition to the C4→C8 bond, a second linkage takes place in the form of a C2→C7 ether bond [16]. In grape seed, only B-type procyanidins (Figure 2) have been conclusively reported, although a single study suggests the presence of A-type procyanidins in a white *Vitis vinifera* variety (‘chardonnay’) based on tandem mass spectrometry analyses [44].

Among the 17 dimeric procyanidins reported in grape seeds [28,29,45,46,47,48,49], nine are esterified to one or two gallic acid moieties, mostly attached to an (−)-epicatechin unit (Figure 2), with a high level of selectivity suggestive of enzymatic esterification. Only one dimer had gallic acid esterified onto a (+)-catechin unit. The most common and abundant procyanidins found in grape seeds are the C4→C8 linked dimers (procyanidins B1 to B4, Figure 2), while comparatively less C4→C6 linked isomers (procyanidins B5 to B8, Figure 2) have been reported [28,29].

Procyanidin B2 (epicatechin-(4β→8)-epicatechin, Figure 2) is consistently the most abundant procyanidin dimer in grape seeds, followed by the other C4→C8 linked dimers and their 3-*O*-gallates [27,29,47]. Interestingly, while 3-*O*-gallate derivatives of procyanidins B1, B2, B4, B5, and B7 (Figure 2) are commonly reported, there is only one report of procyanidins B3- and B6-3-*O*-gallates [46] in grape seeds and no reports of gallate derivatives of procyanidin B8. The report of procyanidins B3 and B6 gallates in *V. vinifera* seeds by de Freitas et al. [46] was based on comprehensive methods that included mass spectrometry experiments, enzymatic hydrolysis, partial thiolysis, and chromatographic comparisons with authentic standards of monomer units. However, other studies corroborating these findings are not yet published. It is currently unknown why O-gallates of procyanidins B3 and B6 are uncommon or even potentially absent in grape seeds.

Among the 16 trimeric procyanidins reported in grape seeds [28,29,45,46,50], procyanidin C1 and epicatechin-(4β→8)-epicatechin-(4β→8)-catechin (Figure 3) are usually the most abundant trimers [29]. Four different backbones (A–D: Figure 3 and Figure 4) are responsible for the diversity of all known C-type procyanidins, which differ by the linking position of their monomer units. Among them, backbone A (Figure 3) has the highest structural diversity of procyanidin trimers. In this structural class, the three monomer units are linked through alpha or beta bonds in positions C4→C8 (Figure 3). The C4→C6 link in C-type procyanidins is rare, with only one trimer known (backbone D: Figure 4), which was tentatively reported in grape seeds by de Freitas et al. [46].

Backbone types can include different combinations of catechin and epicatechin as the initiation, extension (middle), or terminal subunits. Downey et al. [51] reported that there is no obvious reason for selection of specific flavan-3-ols as terminal and extension subunits of procyanidins [51], yet several studies contradict this [17,29,30,46,52]. Generally, the dominant extension unit is (−)-epicatechin, even when (+)-catechin is more abundant in the extract [29]. However, as the relative amounts of epicatechin increase, the occurrence of this monomer in other positions also increases. According to Santos-Buelga [29], “the seeds with higher percentages of (−)-epicatechin also show higher relative proportions of procyanidins containing this compound in the initiation unit and, likewise, the higher relative percentages of procyanidins with (+)-catechin in their terminal unit are generally related to elevated proportions of this monomer.”

Out of the 16 procyanidin trimers reported in grape seeds, ten have (−)-epicatechin as the extension unit, four have (+)-catechin, and two have (−)-epicatechin-3-*O*-gallate (Figure 3 and Figure 4). (−)-Epicatechin is also more common as the initiation unit, comprising 12 out of the 16 when compared to just the four of (+)-catechin. (−)-Epicatechin is also more commonly galloylated, i.e., among the 33 dimeric and trimeric procyanidins currently elucidated (Figure 2, Figure 3 and Figure 4), only three show galloylation on a (+)-catechin unit, while twelve show galloylation on (−)-epicatechin units. No trimers have been found containing gallate esterifications in the initiation unit, even though they are found in dimers or as monomers. According to Santos-Buelga [29], this could reflect steric hindrance at the active site of esterifying enzymes.

Hence, (+)-catechin is the most abundant terminal unit among procyanidin trimers, with six out of 16, followed by (−)-epicatechin, with five, and (−)-epicatechin-3-*O*-gallate, with four. This suggests that the initiation units come from different metabolic pools than the extension units, as previously hypothesised [53,54]. The fact that none of the larger oligomers (4+ units) include both (+)-catechin and (−)-epicatechin in the extension section also supports this. However, it is too early to comment on the validity of this hypothesis because very little is known about procyanidin oligomerisation.

Aside from the known dimers and trimers, much higher degrees of polymerisation also occur. For instance, procyanidins with a degree of polymerisation from 28–37 units and a degree of galloylation from 8–15 have been observed [17,55]. Furthermore, Sica et al. [19] demonstrated that some tannins in grape seeds are as large as >100,000 Da. The complete structures of these large polymers are not yet determined, as the binding positions and identity of each of the extension units needs to be clarified [19]. Mass spectrometry experiments convey that there are many of these polymers awaiting full characterisation. From what little is known, the degree of galloylation decreases as the degree of polymerisation increases [56], and a biosynthetic explanation is lacking. As the galloylation of polymeric procyanidins seems to occur preferably on (−)-epicatechin, a high specificity of the esterifying enzymes is likely [29] and the steric hindrance of polymers may be exclusive, as previously mentioned.

In conclusion, despite recent advances there are still significant gaps in our knowledge about the chemistry of grape seed flavan-3-ols. For example, the absolute structural characterisation of glycosylated flavan-3-ols and oligomeric procyanidins, the presence of galloylated monomers in seeds, and the biosynthetic steps necessary for their formation is still lacking. This information is important because monomeric and polymeric flavan-3-ols are involved in the biological and nutritional properties attributed to grape seed extracts.

#### 2.1.2. Biosynthesis

Even though the mechanism of flavan-3-ol polymerisation is yet to be described, the biosynthesis of the major flavan-3-ol units in grape seeds, (+)-catechin and (−)-epicatechin, is well-known. The main enzymes and intermediates leading to the formation of flavan-3-ol monomers are depicted in Figure 5. However, more elaborate details are provided in other studies [5,57,58].

The biosynthesis of flavan-3-ol monomers in grapes follows the same initial steps of other flavonoids via the phenylpropanoid pathway, leading to the production of narigenin by chalcone synthase (CHS) and chalcone isomerase (CHI). Narigenin is oxidised by flavanone 3-hydroxylase (F3H) followed by either flavanone 3′-(F3′H) or 3′,5′-hydroxylase (F3′5′H). The latter two enzymes create two possible flavan-3-ol precursors: dihydroquercetin for catechin and epicatechin (by F3′H); or dihydromyricetin for epigallo- and gallocatechins (by F3′5′H). Thereafter, dihydroflavonol 4-reductase (DFR) converts the precursors to leucoanthocyanidins, including leucocyanidins (from dihydroquercetin) and leucodelphinidins (from dihydromyricetin) (Figure 5). In the final step, the leucoanthocyanidins are converted to a flavan-3-ol, either by leucoanthocyanidin reductase (LAR), giving catechin or gallocatechin, or by the combination of enzymes leucoanthocyanidin dioxygenase (LDOX) and anthocyanidin reductase (ANR) to create their epimers, epicatechin and epigallocatechin, respectively (Figure 5).

Considering that prodelphinidins are exclusive to the skin of the grape berry, the enzyme F3′5′H is expected to be active in this site. Studies on the temporal and tissue-specific expression of genes encoding F3′H, F3′5′H in grapes demonstrated that these two enzymes, in addition to having a cytochrome b5 (CytoB5) isotype, are necessary for the hydroxylation of the B ring of flavan-3-ols to produce (epi)gallocatechin [59]. According to Bogs et al. [59], F3′H and CytoB5 but not F3′5′H were expressed in grape seeds, consistent with the accumulation of 3′-hydroxylated flavan-3-ols (catechins) in this organ. In the skin of the grape berry, all three genes were expressed, particularly after the onset of ripening (veraison), and their expression correlated well with the accumulation of 3′- and 3′,5′-hydroxylated flavan-3-ols (catechins and gallocatechins, respectively) [59]. Similar results were obtained by Jeong et al. [60], who described a higher expression of F3′H and accumulation of 3′-hydroxylated flavan-3-ols (catechins) in the seeds, while a higher expression of F3′5′H and accumulation of 3′,5′-hydroxylated flavanols (gallocatechins) was observed in the skins [60]. These observations corroborate the observed spatial distribution of the different types of flavan-3-ol monomers.

Regarding the temporal regulation of these genes, Bogs et al. [59] found that the expression of F3′H was highest before flowering, when 3′-hydroxylated flavonols are biosynthesised, while CytoB5 and F3′5′H were highly expressed after flowering, when prodelphinidins are mainly accumulated. Furthermore, “in contrast to red grapes, where F3′H, F3′5′H, and CytoB5 were highly expressed during ripening, the expression of F3′5′H and CytoB5 in white grapes during ripening was extremely low, suggesting a difference in transcriptional regulation” across different grape varieties [59]. Since the enzyme F3′5′H and its product (dihydromyricetin) are also involved in the biosynthesis of delphinidin-type anthocyanins responsible for red to purple colours, this observation makes sense.

Despite considerable progress in the understanding of the flavan-3-ol biosynthetic pathway, the mechanism of flavan-3-ol polymerisation remains unknown. Furthermore, flavonoid synthesis occurs in the cytoplasm, yet most products are stored in specific compartments, such as cell walls and vacuoles. Hence, an efficient, albeit unknown, transport mechanism exists [5]. Zhao and Dixon [61] reported the expression of a glucosyltransferase enzyme (UGT72L1) in the seed coat of *Medicago truncatula* (Fabaceae), concluding that MATE1 is an essential membrane transporter for procyanidin biosynthesis in *Medicago* [61]. The over-expression of this enzyme involved in the biosynthesis of epicatechin 3′-*O*-glucoside results in increased accumulation of procyanidins [62]. Further studies have suggested that epicatechin 3′-*O*-glucoside rather than free epicatechin is the substrate for procyanidin biosynthesis acting as intermediate in the polymerisation of flavan-3-ols in *M. truncatula* [63]. A similar mechanism is believed to occur in grape seeds. The presence of epicatechin 3′-*O*-glucoside in the seeds of different grape varieties further supports this hypothesis [33,41]. However, a recent study by Liu et al. [64], suggest a possible role of the enzyme LAR in the regulation of the biosynthesis of oligomeric proanthocyanidins [64]. This study, focused on LAR mutants of *M. truncatula,* showed that a loss of function of this enzyme leads to the accumulation of 4β-(S-cysteinyl)-epicatechin, which provides the C4→C8-linked extension units during non-enzymatic procyanidin polymerisation, resulting in increased levels of oligomeric procyanidins.

#### 2.1.3. Pharmacokinetics and Structure–Activity Relationships

In vitro assays that elucidate important structural features of procyanidins often fail to predict in vivo outcomes. This is because of the unforeseen challenges of natural product pharmacokinetics. For example, it is common to ascribe biological effects to metabolites at concentrations that are many orders of magnitude above the feasible plasma concentrations [65]. Furthermore, metabolites that enter the body via digestion are often reduced by removal of O- and COO-linked moieties in the alimentary canal, then transformed by conjugation in Phase 2 transferase activities at the liver and in the body’s tissues by different enzymes. Alternatively, glycosylated structures, such as anthocyanins, are absorbed early in the alimentary canal (the small intestine) and have very short half-lives, due to higher aqueous solubility and, hence, poor affinity for blood plasma proteins, giving rapid elimination from the body (renal elimination) [65]. Furthermore, certain flavonols show low gastrointestinal absoprtion rates and poor solubility in aqueous media and, therefore, their in vivo outcomes are limited, albeit their in vitro potential.

Polyphenols are less likely to be oxidised in Phase 1 reactions because the molecule is already highly oxidised. Rather, polyphenols tend to be circulated around the body as xenobiotic conjugates, either as sulphate esters or glucuronides [66]. Cushnie and Lamb [67] discuss the effects of conjugated polyphenols in blood serum and point out that although renal elimination is the end-purpose of these Phase 2 reactions, conjugated xenobiotics may also participate in other highly specific therapeutic effects in infected or inflamed tissues [67]. In mammals, infected tissues secrete β-glucuronidases and other enzymes that cleave hydrophilic moieties from conjugated xenobiotics and significantly reduce their blood plasma solubility. This leads to an accumulation of polyphenols in specific unhealthy tissues, raising local concentrations to therapeutic levels comparable to those measured in vitro. Because polyphenols are generally not oxidised in metabolism, the reducing enzymes return xenobiotics to pre-conjugated forms. This means that in vitro assays of condensed tannins (procyanidin oligomers and polymers) and catechin monomers (catechin and epicatechin) are feasible in theory. In this regard, it has been observed that repeated feeding of mice with grape seed extracts leads to an accumulation of gallic acid, catechin, and epicatechin in brain tissues [68]. This may have positive implications for chronic or terminal cerebral afflictions, such as Alzheimer’s disease or brain cancers.

In vitro assays often demonstrate that the most potent effects are derived from galloyl derivatives (gallic acid esters) and C4→C6 linked isomers. For example, procyanidin isomers with a C4→C6 linkage showed stronger epidermal lipid peroxidation inhibitory activity than C4→C8 isomers [49], which may be related to the reduced steric hindrance of C4→C6 linked dimers. Lipid peroxidation is also strongly inhibited by gallic acid esters of procyanidin, particularly when esterified to the 3′-hydroxy position of a procyanidin dimer [49]. Unfortunately, gallic acid esters (gallates) are the least effectively absorbed in digestion and they tend to be rapidly methoxylated in metabolism [66] or cleaved at the ester bond, altering their biological activity [69]. However, gallates are not modified at the body’s earlier metabolism sites, such as the mouth, oesophagus, and stomach. When gallates enter the mouth, they influence the aesthetics of procyanidin rich foods and drinks via the precipitation of salivary proteins. Hence, sensory perception of astringency depends mainly on the degree of galloylation in the extract [37,70]. This is also true for so-called gallates that are not esters, such as epigallocatechin, which has been shown to inversely affect the astringency of wine [71]. Because epigallocatechin is characterised by a hydroxylation pattern on ring B that resembles the major fragment of gallic acid, this demonstrates that the trihydroxybenzoyl moiety is the structural requirement, and not the ester *per se*, for reducing the perception of astringency. Other key structural characteristics such as the mean degree of polymerisation, the conformation, and hydrophobicity of the procyanidins are also important because they influence their interaction with macromolecules [72].

### 2.2. The Broader Grape Seed Metabolome

The other phenols in grape seed include phenolic acids, stilbenes, hydrolysable tannins, and flavonols (Figure 6). These are usually present only as minor components. Among phenolic acids, gallic acid is the most common in some varieties such as ‘chardonnay’ and ‘pinot blanc’ [73,74,75], but concentrations are variable. Other phenolic acids like protocatechuic acid are more consistent across varieties [76]. Phenolic acids are also present as glycosylated derivatives, but proper structural characterisation is still required [77,78].

While stilbenes are common in the fruit pulp and grape skins [79], they seem to occur as mere trace ingredients in the seeds (Figure 6) [80,81,82]. *Trans*-resveratrol is the most famous stilbene in grape skin and is widely implicated in positive health outcomes. Some studies report traces of both *cis*- and *trans*-resveratrol isomers in seeds [81,82], but results are inconsistent [34,83]. Since the isomerisation of *trans*-resveratrol into its *cis* analogue has been shown to occur by natural light [84], it is recommended that studies reinvestigate its natural occurrence in grape seeds by carrying out its extraction in the dark [79]. A recent study that used more a rigorous methodology confirmed earlier findings of resveratrol in seeds [85], but confirmatory studies are still necessary. Other studies have also detected new stilbenes in grape seeds that were tentatively assigned as *cis* and *trans* isomers of piceid, piceatannol, and miyabenol C [86].

An investigation of hydrolysable tannins was performed by Sandhu and Gu [77]. They identified 33 from two main groups: (1) gallotannins, and (2) ellagitannins. The grape species with the highest number of reports of hydrolysable tannins in seeds is *V. rotundifolia*, or ‘muscadine’ grape [75,77,87]. Nevertheless, Prodanov et al. [75] also identified hydrolysable tannins and ellagic acid glycosides in seeds of the ‘malvar’ cultivar of *V. vinifera*. However, it may be necessary to exclude the possibility of metabolite transfer from other plant parts. For example, studies that utilise grape seeds as by-products from the wine industry, such as the studies by Garcia-Jares et al. [73] and Pasini et al. [74] may need confirmation, since the processing technique (i.e., entire grape maceration) can persuade diffusion of metabolites from other plant parts. The recent report of anthocyanins in grape seeds [12,85], a chemical class responsible for the characteristic red and purple colours of grape skin, highlights this issue.

Contrary to hydrolysable tannins, which have a restricted distribution across species and cultivars, ellagic acid [75,77], quercetin [73], quercetin 3-*O*-rhamnoside [77], quercetin-3-*O*-glucoside [73,85], quercetin-3-*O*-glucuronide [73], rutin [19], kaempferol and myricetin [87] seem to be ubiquitously expressed (Figure 6). Other ellagic acid and flavonoid glycosides (structures not shown) have also been assigned tentatively by interpretation of LC-MS/MS data [74,75,77,78,88].

In addition to the phenolic compounds, which usually represent between 4 and 7% of the seed weight, grape seeds demonstrate a complex metabolome containing 11% protein, 35% fibre, 16% oil, 3% minerals, and a fraction of other metabolites including free amino acids and simple organic acids [25,89]. The main amino acids include tryptophan, glutamic acid, proline, and tyrosine [19,75]. The main organic acids are malic acid, citric acid, and succinic acid [19]. Lastly, additional organic acids and amino acids have also been reported in ‘cabernet-sauvignon’ grape seeds using untargeted metabolomics approaches [12].

## 3. Modulating Factors

Several studies have shown that the content of phenolic compounds in grapes and their seeds is influenced by both extrinsic and intrinsic factors. Intrinsically, the variety or species is evidently significant. Extrinsically, the environmental characteristics such as solar radiation, water availability, ambient temperature cycles, location (altitude), stage of ripeness, and viticultural practices, are also important [51,90,91,92,93,94,95,96,97]. However, as these works have been conducted in different research groups under different conditions, the data collected are sometimes conflicting. Consequently, the relative contribution of each factor to the chemical variation of grape seed remains elusive.

The ideology behind understanding modulating factors is to harness some degree of control to optimise expression of phenolics for their health or aesthetic benefits. One of the first studies comparing the influence of different factors on the accumulation of highly polymerised flavan-3-ols in grape seeds [56] concluded that the accumulation of these compounds is primarily driven by genetic factors (represented by the grape variety), while climatic conditions, represented by the sampling year, had a less significant influence. However, it is possible that only some types of metabolites can be modulated this way and the individual contribution of genetic and climate factors on the chemistry of grape seed is still far from being accurately elucidated. Aspects of the modulating factors can be categorised as (1) intrinsic, focused on genetics in species or varieties, (2) extrinsic environmental, focused on physical conditions, and (3) extrinsic managerial, focused on developmental stages and optimal harvest times.

### 3.1. Genetic Factors

Although the concentration and composition differences in grape seed flavan-3-ols can be influenced by external factors, genetic divergence within the species constitutes a major driving force of chemical differentiation in grapes [56]. Furthermore, studies suggest that flavan-3-ols are expressed differently according to cultivar, when subjected to the same extrinsic environmental factors [34,77,87,98]. Interestingly, as the genetic relationships become increasingly distant, chemical expression patterns also become more distant, with the greatest chemical divergence demonstrated by interspecific (species-level) differences.

#### 3.1.1. Species-Level (Interspecific) Differences

While most chemical studies of grape seeds in the published literature are based on *V. vinifera*, there are important qualitative and quantitative differences in the phenolic composition of different grape species. Polyphenols in the seeds of 91 grape accessions from 17 species in *Vitis* demonstrated significant interspecific differences in the concentration of monomeric flavan-3-ols, procyanidins B1 and B2, flavonols, isoflavones, and gallic acid derivatives [99].

From the 17 species, *V. palmata* yielded the highest content of total polyphenols, giving 21.02 mg/g (fresh weight, FW), followed by *V. vinifera* (17.63 mg/g, FW) and *V. vulpina* (19.49 mg/g FW) [99]. These three species yielded significantly higher than the others, with the lowest content in *V. champinii* (0.95 mg g−1 FW). A similar pattern was observed in the specific analysis of monomeric flavan-3-ols, dimeric and trimeric procyanidins, and flavonols, reiterating that *V. palmata*, *V. vinifera,* and *V. vulpina* provide the highest natural sources of these compounds, except for resveratrol, which was abundant only in *V. monticola* [99].

While *V. vinifera* is the most cultivated species of grape, other species such as *V. labrusca* and *V. rotundifolia* (muscadine grapes), are especially popular in the Eastern and South-eastern United States. By comparison with *V. vinifera*, these species express less (+)-catechin and (−)-epicatechin in the seeds, but more gallic acid and ellagic acid [34,87,100]. For instance, ellagic acid, its glycosides, and ellagitannins can be used as chemical markers of *V. rotundifolia* [77,87]. Furthermore, the structural diversity of hydrolysable tannins (gallotannins and ellagitannins) in *V. rotundifolia* surpasses other species [77], particularly *V. vinifera*, which expresses only trace amounts of tannins [29].

The presence of hydrolysable tannins is a consistent feature of *V. rotundifolia*. Studies with different cultivars of *V. rotundifolia* grown in different places with potentially different climate conditions such as Central Florida, USA [77], North-western Florida [87,98] and South China [87], have consistently shown the presence of hydrolysable tannins (mainly ellagitannins) in the seeds of this species. However, the concentration of these compounds and other phenols seems to vary significantly according to the variety of muscadine grape and their growth location. The seeds of red cultivars grown in Florida, USA, and Nanning, China exhibited higher contents of ellagic acids and precursors than those in Pu’er-Yunnan, China, while the concentration of seed flavonols was almost doubled among grape cultivars grown in the USA when compared to those in China [87].

In this regard, there is minimal research dedicated to understanding the degree of polymerisation and galloylation of procyanidins across the different species. This has significant implications in the context of health outcomes [22,101]. Although procyanidin dimers and trimers have an absorption rate less than 10% of (−)-epicatechin, they have significant impacts on health. As the degree of polymerisation increases, the absorption rate decreases until absorption stops at five or six units [22]. Hence, cultivars that express optimal oligomer sizes could be used as gene pools in cross pollination strategies with *V. vinifera* to develop high-yielding, biologically active grape seed extracts [99].

#### 3.1.2. Variety-Level Differences

Multivariate statistical methods strongly emphasise the importance of genotype in determining the chemical profile of a specific cultivar [56]. Previous studies based on principal component analysis (PCA) of flavan-3-ol monomers showed a clear grouping pattern by genotype or variety, even when samples were collected in different locations [30]. Another study of seventy different varieties, both red and white, demonstrated that sampling from different years did not affect the chemical agreement between genotypes [36]. The variety ‘pinot noir’ had the greatest levels of flavan-3-ols, being 30-fold greater than the lowest, ‘kerner’ [36]. The high content of flavan-3-ol from ‘pinot noir’ seeds is supported by multiple studies [30,47,99].

In any given year, the species’ genotype is the most significant determinant of a seed’s chemical profile. However, if specimens from the same cultivar face differences in extrinsic factors, chemical profiles diverge significantly. Chira et al. [102], observed this in a study of ‘cabernet-sauvignon’ and ‘merlot’ over two consecutive years. They found that the two varieties were clearly different in chemical profiles, and the chemistry of each variety changed over consecutive years [102]. Furthermore, studies suggest that the chemistry of some grape varieties are more plastic than others. Specifically, seeds from ‘Albalonga’ and ‘tempranillo’ had very different chemical profiles in different years, but this was not the case for seeds from ‘Bacchus W.’, and ‘graciano’ [36,56]. Similarly, Pérez-Navarro et al. [42] demonstrated that ‘tempranillo’ grapes did not vary by year, whereas the ‘tinto fragoso’ variety did [42].

In general, chemical differences between red and white varieties are observed in the proportion of galloylated and non-galloylated procyanidin dimers [47]. One study reported that red grape varieties contained more monomeric flavan-3-ols and procyanidin dimers than seeds of white grape varieties [47]. However, other studies reported a much lower proportion of monomers in red grapes [29,103]. These inconsistencies are common in the published literature and may very well be a consequence of stronger differences in extrinsic factors, i.e., drought versus rainy years.

Generally, across varieties, the relative quantity of monomers to oligomers varies, as well as the degree of galloylation [76,103] and the relative amounts of monomeric catechin when compared to epicatechin [103,104,105]. White most grape varieties tend to express more consistent levels of (+)-catechin when compared to (−)-epicatechin [73], some varieties express significantly higher levels of galloyl derivatives, such as the two red varieties ‘merlot’ and ‘carménère’ [29,30,47,76]. In contrast, the identity of extension and terminal units in oligomers tend to be consistent across varieties [103]. Generally terminal units are (+)-catechin, and extension units are overwhelmingly (−)-epicatechin. This consistency is somewhat independent of the ratio differences and yield of the monomeric forms mentioned above.

### 3.2. Environmental Factors

The impact of the growing environment on grapes has been recognised for centuries; the concept of ‘terroir’ was developed to describe the unique influence of a region’s soil type, temperature, rainfall, and other environmental factors on the characteristics of the grapes it produces. Additionally, agronomic strategies such as alteration of environmental conditions (light, temperature, mineral nutrition, and water management), application of elicitors, stimulating agents, and plant activators have been employed to enhance the biosynthesis of polyphenols in grape [77,94,106,107]. However, as most of these studies have focused on evaluating the impact of different factors on the chemical composition of the entire berry or only its skins, the influence of those factors on the seed chemistry have not been as comprehensively explored. Nevertheless, details can be realised from annual studies of common vineyards, i.e., two-way ANOVA has revealed that the higher polymerised procyanidins are more plastic (influenced by extrinsic factors) than less polymerised ones; non-galloylated procyanidins are highly plastic, yet mono-galloylated procyanidins are not [56]. Unfortunately, the influence of more specific climatic factors such as UV radiation, temperature, precipitation, and soil type have not been comprehensively studied.

Solar radiation has long been known to influence the development of the grape berry and the biosynthesis of phenolic compounds, which ultimately affect wine quality [94]. However, the effect of solar radiation on the accumulation of flavan-3-ols in the seeds of the grape berry has not been explored in detail. For example, when shading cloths were used on ‘pinot noir’ grapes, there was no significant effect on the procyanidin content of seeds, but the ratio of (−)-epicatechin monomers increased. In tannins, the frequency of (+)-catechin as both a terminal and extension subunit slightly increased, and so did the mean degree of polymerisation (mDP) [108]. However, this effect of shading on terminal subunit composition was not observed in another study [109]. Furthermore, an investigation of the effects of UV-B radiation exclusion was inconclusive because seed chemistry demonstrated greater variation by year [91]. The same authors noted that the light intensity may be critical during the flowering stage.

Although studies have remained inconclusive about the effects of light manipulation on the accumulation of seed flavan-3-ols, it is well known that sunlight *per se* triggers the expression of key genes, such as phenylalanine ammonia lyase (PAL) and the enzyme chalcone synthase (CHS), which are essential for flavonoid biosynthesis [110]. A recent study observed differences in the transcriptome by altering sunlight exposure, which changed the expression of other types of phenolics in fruit pulp, while the effects to the expression of flavan-3-ols in the skin were inconclusive [111].

Conceptually, changes to light will affect plant surfaces first, but it is unclear how this transmits to the internal organs, such as the seeds in which procyanidins accumulate. A possible hypothesis is that solar radiation works indirectly, by stimulating the biosynthesis of flavan-3-ol precursors such as narigenin or dihydroquercetin (Figure 5). Due to the requirement of these flavonoids to feed flavan-3-ol synthesis, an efficient flavonoid transport system [112] is essential during berry and seed development [5]. Hence, as light or solar radiation modulates flavonoid synthesis in skin and fruit pulp, then flavonoid concentration and transport efficiency modulates flavan-3-ol synthesis in seeds, and then downstream oligomers. This hypothesis is further supported by the fact that “the site of seed flavan-3-ol biosynthesis and storage is known to differ at subcellular, cell, and even tissue levels, meaning that an efficient flavonoid transport system is required all along berry and seed development”, as reported by Rousserie et al. [5].

In a review of the impact of seasonal temperatures on flavan-3-ol biosynthesis, Gouot et al. [113], concluded that there is insufficient research on grapes to draw any conclusions. However, they suggested that the independent biosynthesis of procyanidins in grape seed somewhat shelters the process from abiotic factors. Nevertheless, narrowing the natural diurnal temperature range promoted the ripening of ‘merlot’ grapes and slightly increased the concentration of procyanidins in the seeds [114]. Lower cultivation temperature also led to increased procyanidin content without changing the profile of constituents [115]. The same study also revealed that higher temperatures inhibit the expression of anthocyanidin reductase (ANR) and leucoanthocyanidin reductase-1 (LAR1) in the grapes’ skins, which are key genes in procyanidin biosynthesis. Similarly, artificially elevating temperatures led to a significant decrease of extractable seed tannins [116]. However, it was speculated that higher temperatures promoted ripening of the berry, which reduced the extractability of the tannins, due to binding to cell wall components.

Other studies observed that very hot summers were associated with high quantities of procyanidin B1 in seeds [117]. Conversely, areas with milder summers, such as Navarra (Northern Spain) [103] and Bordeaux (France) [26], expressed more procyanidin B2 in seeds [97]. However, Gouot et al. [113] stress that biosynthetic studies of grape varieties are not controlled enough to draw robust conclusions on the effects of light and temperature. In contrast, the effects of natural or anthropogenic watering regimes are undisputed.

Irrigation of vineyards during the critical developmental stages before ripening is a widespread practice, which has a positive effect on the phenolic content of seeds [118,119]. Conversely, water deficit early in berry development decreased the biosynthesis of flavan-3-ols in the skins [120], but later in berry development causes an increase in the concentration of anthocyanins and total phenols in grape skins [92,121]. While strategic reduction of watering volume seeks to increase flavours of fruits and wines, grape seed chemistry is not significantly impacted, as it occurs late in the seed’s development stages. For example, while the biosynthesis of flavan-3-ols in grape berries was modified when plants were given little to no water [95,122], no effect on procyanidin content of the seeds was observed [95]. Furthermore, multiple studies have confirmed that the composition and concentration of seed tannins is unchanged by water stress [95,123], but additional water might slightly increase the mDP of flavan-3-ol oligomers [124].

Due to the complexity of processes leading to the biosynthesis of grape seed procyanidins, the ideal conditions for optimisation of chemical profiles have not yet been realised. It is not a single factor, but many, that control the expression of procyanidins. Unfortunately, experiments fail to control for all possible variables, i.e., shading treatments not only reduce light, but also affect temperature and humidity, causing changes in transpiration and other factors [94]. Temperature and light penetration may cause reciprocal effects. For example, solar radiation drives the expression of genes involved in early procyanidin biosynthesis (e.g., PAL and CHS) [5], while higher temperatures inhibit the expression of genes involved in latter steps (ANR and LAR1) [115]. Thus, research strategies need to incorporate a more complete understanding into experimental designs to attain a higher degree of control and increase the reproducibility of study outcomes.

### 3.3. Developmental Factors

Grape seeds develop over three phases. In Phase I, (histodifferentiation) the seed is fertilised, and cells divide rapidly, forming all seed structures paralleling the creation of the immature fruit. In Phase II (expansion/reserve disposition), reserve materials are accumulated, and no further cell division takes place. This phase is associated with veraison, and towards the end of this phase the vascular connections between the seed and the plant close. Finally, in Phase III (maturation/drying), the seeds mature by desiccation [5,125].

Over the course of the seed and berry’s ripening, the profiles of flavan-3-ols similarly go through phases. In Phases I and II, the concentration of seed flavan-3-ols increase but diminish as the fruit ripens (Phase III) [50,90,91,126,127]. Flavan-3-ol accumulation is concomitant with the expression of genes encoding ANR and LAR enzymes [59].

Procyanidins accumulate at a stage different to flavan-3-ol monomers. Procyanidins accumulate very early during seed development [126,127], while monomers increase sharply toward veraison. At this point, procyanidins and monomers peak and start declining thereafter [50,76,96]. Allegedly, procyanidin dimers decline at a slower rate than monomers [50], creating a change in the flavan-3-ol profiles after veraison [126,127]. However, it is possible that chemical profile changes are a consequence of lower extractability, due to changed oxidative states of molecules, rather than from degradation [51,126,127]. Kennedy et al. [126,127] found evidence of this using electron paramagnetic resonance (EPR) spectroscopy. By following the formation of radical species in the developing seeds, they observed low radical concentrations until a sudden increase at veraison, which peaked three weeks later [126,127]. Because radicals oxidatively modify polyphenols, their extractability may decrease.

Based on their work, the dynamics of flavan-3ol and procyanidin accumulation in grape seeds were classified into four distinct stages:Procyanidin biosynthesis, peaking during rapid growth of juvenile berry (Phase I);Flavan-3-ol monomer biosynthesis: as the biosynthesis of procyanidins slows down, monomers increase (Phase II). Kennedy et al. [126,127] explained that the overall rate of biosynthesis remains constant, which implies a rate limiting step, probably the common precursor leucocyanidin. This phase ends at the end of Phase II;Programmed oxidation: At veraison and after monomer biosynthesis declines. Monomers and procyanidins are oxidised (visible as a browning and hardening of the seed coat);Non-programmed oxidation: the seed is fully desiccated. Little change in extracted polyphenols occurs during this phase.

Interestingly, glycosylated forms of flavan-3-ol monomers do not correlate to concentration changes of free flavan-3-ols and procyanidins. Even after Phase II, their relative concentration continues to increase and peak at Phase III [33]. There are several other metabolite accumulation patterns that have not been consistent across studies, such as the increase in mDP, when grapes reach over-ripeness [51,90,91,96]. However, the results by Kennedy et al. [126,127] are especially reliable.

### 3.4. Other Factors

As previously mentioned, it is problematic to discuss a single abiotic factor because such factors are usually linked to other effects. For example, linking an aesthetic improvement of wines to the physical location and elevation of the vineyard is not reliable because there may also be rainfall differences, among others. Even local microclimates can create chemical variation, i.e., a study exploring the chemical composition of the ‘albariño’ variety of white grape, cultivated in different vineyards restricted to the region of Galicia (Spain), found that flavan-3-ol profiles varied widely. Seeds from grapes grown in the Ribeira Sacra region contained higher concentrations (up to double) of flavan-3-ols than those grown in other regions, such as O Rosal [73].

While difficult to validate findings related to elevation, patterns related to altitude have been observed. A Turkish cultivar (Ekşikara), grown at two different altitudes of 1000 m and 1500 m [128], demonstrated a higher monomeric and dimeric flavan-3-ol content in the seeds at higher altitudes. Similarly, ‘syrah’ grapes in Brazil expressed higher concentrations of condensed tannins and galloylated procyanidins at lower elevations (350 m) and higher amounts of monomers and dimers at higher altitudes (1100 m) [93]. However, as previously mentioned, it is exceedingly difficult to account for all the variables, such as humidity, rainfall, and temperature, at different altitudes.

## 4. Quality and Adulteration

Catechins and procyanidin-rich grape seed extracts started appearing in the market as nutraceuticals in the early 1990s [47]. Over the last 25+ years, this initiative has grown into a multi-million-dollar industry. However, the inconsistency of grape seed chemistry has raised concerns about the reproducibility of studies that demonstrate safety [18]. The United States Food and Drug Administration (FDA) classified grape seed extracts as “generally recognised as safe (GRAS)”. A dose of 1.78 g/kg body weight/day in male rats and 2.15 g/kg body weight/day in female rats showed no adverse effects [129]. Further studies revealed that the dose required to induce a 50% mortality (LD_50_) is higher than 4 g/kg in rats [130]. In humans, oral intake of grape seed extracts up to 400 mg for 12 weeks and 2500 mg for 4 weeks were considered safe in a study of 61 healthy individuals [131].

The quality and efficacy of commercial grape seed products depends upon the chemical composition of monomeric catechins and oligomeric procyanidins. While grape seed procyanidins are non-specifically listed as the active ingredient in dietary supplements, the most important are the oligomeric procyanidins [19] because they have greater bioavailability than larger polymers [22,66,68]. As previously mentioned, only degrees of polymerisation lower than five are absorbed, and this is much slower than monomers such as (−)-epicatechin [22].

The degree of polymerisation is not the only factor influencing absorption or bioactivity. As previously mentioned, the degree of gallate esterifications on procyanidin subunits, their conformation, and hydrophobicity are also important [37,49,70,72,101]. The current standardised criteria of quality as prescribed by the United States Pharmacopeia of “no more than 19.0% of catechin and epicatechin” [18] is very broad. Furthermore, no guidelines are in place to detect adulteration with other plants such as pine bark or peanut extracts that are rich in A-type procyanidins. Yet, A-type procyanidins can be easily detected for authentication purposes [18,19] because the linkage position of individual monomers in oligomeric procyanidins can be deduced from mass spectral fragmentation patterns or by other spectroscopic methods (i.e., NMR).

Single linkages between C4→C6 or C4→C8 give rise to the characteristic B-type proanthocyanidins often found in grape seed, whereas A-type proanthocyanidins have the same C4→C8 bond, but with a second linkage in the form of a C2→*O*→C7 or C2→*O*→C5 bond (Figure 7). The absence of these metabolites in grape seed has been widely recognised by the scientific community [18,19]. Unfortunately, the differentiation between grape seed procyanidins from those originating from other sources (e.g., peanut skin and pine bark) is an important quality criterion often overlooked by regulators. The absence of prodelphinidins (derived from the skin of the grape berry) is also an important consideration in the quality assessment of grape seed extracts [43].

Adulation of grape seed-based dietary supplements is still a common problem. A recent survey of 21 commercial grape seed products concluded that nine of these supplements (43%) were adulterated with peanut skin extracts and six samples (28%) were determined to be devoid of grape seed extract [18]. Furthermore, pine bark extracts demonstrate a strong chromatographic resemblance to grape seed, especially according to the flavan-3-ol monomers and dimers. It is likely that adulteration with pine bark escapes the current phytochemical authentication checks [18]. This is unfortunate, as grape seed and pine bark extracts differ in their bioactivity [101]. An additional issue is that the price of extracts does not correlate with their quality. Unfortunately, according to a previous study, “consumers are paying arbitrary prices, not reflective of the quality of commercial products and their associated label claims” [18].

The problems of adulteration in grape seed extracts are recognised as a serious concern by the American Botanical Council in their Botanical Adulteration Program [132]. The most common adulterants are peanut skin and pine bark extracts, as they also contain the same (epi)catechin and procyanidins [18]. However, the simple analytical techniques typically used in industry and suggested by the regulators (e.g., colorimetric assays and thin layer chromatography) cannot clearly identify when grape seed extracts are adulterated. Adulterated products still comply with the USP specification of “no more than 19% of catechin and epicatechin”. However, more sophisticated techniques, such as HPLC-UV comparison with reference standards, LC-MS, and NMR, can readily determine whether adulteration has taken place. Therefore, we suggest the following two points should be considered to ensure the quality, authenticity, and efficacy of commercial products:
Check for adulterants: The adulteration of grape seed products with peanut skin and pine bark extracts can be easily distinguished by the presence of A-type procyanidins. These metabolites can be identified by target LC-MS analyses. While procyanidin B2, the main flavan-3-ol dimer found in grape seed extract, shows a deprotonated molecule at *m*/*z* 577.1352, its A-type analogue (procyanidin A2) gives a deprotonated molecule at *m*/*z* 575.1195. The fragmentation pattern of these molecules also shows significant differences that can be used to discriminate between both isomers (Figure 8). The presence of a characteristic fragment ion at *m*/*z* 423.072 (negative ion mode) on the MS^2^ spectrum, resulting from a Retro-Diels-Alder fragmentation, indicates the presence of A-type procyanidins, while B-type procyanidins show a fragment ion at *m*/*z* 425.088 via the same mechanism (Figure 8). The presence of prodelphinidins in grape seed extracts can also be identified by MS analyses. In LC-MS analyses, deprotonated ions at *m*/*z* 609.1249 or *m*/*z* 593.1300 indicate the presence of B-type prodelphinidins, which represent common metabolites in the skins of the grape berry;Quantify bioactive metabolites: As previously noted, grape seed demonstrates complex chemistry, characterised by the presence of tens to hundreds of metabolites at different concentrations. Evidently, the monomeric and dimeric flavan-3-ols (especially catechin, epicatechin, and procyanidin B2) are the main bioactive constituents. Ideally, these metabolites are quantified to give an actual oral dose, particularly of the ingredients that are absorbed [89]. According to Villani et al. [18], out of three authentic extracts, the average total content of the major procyanidins is 383.5 ± 21.13 mg/g, with procyanidin B2 making up 41%. Out of total phenolics, catechin and epicatechin cumulatively representing 32% of the total content detected by HPLC. Unfortunately, Villani et al. [18] demonstrated that several commercial samples have less than 10 mg/g total procyanidins. Some of the extracts did not have any of the three major flavan-3-ols: catechin, epicatechin, and procyanidin B2 [18]. It is therefore necessary that the quantification of these three flavanols is a recognised undertaking in quality control or authentication. By HPLC, the two monomers and procyanidin B2 should cumulatively be above 50% of detected components, and the absolute content of procyanidin B2 should ideally be 150 mg/g of extract.

In addition to the issues of adulteration and safety, the sale of poor quality extracts for use in nutraceutical supplements or in cosmetic products undermines the potential benefits of quality plant-derived products [19].

## 5. Conclusions

Despite the popularity of grape seed-based nutraceuticals, there is more to learn on the chemistry. Since the first reports in the late 1960s to the current date, 45 flavan-3-ols have been unambiguously characterised in the seeds of different grape species and varieties. However, the real number of grape seed flavan-3-ols is likely to be double or triple of this figure. There is a high diversity of glycosylated catechins and polymeric procyanidins that has not been elucidated. Furthermore, many of the chemical studies in the published literature do not comprehensively characterise components, with assignments stopping at the level of class or subclass. Unfortunately, greater specificity is required to make predictions of quality, absorption, and health benefits. Furthermore, when metabolites are identified in the seed extract that originate from other fruit parts, such as the skin (e.g., prodelphinidins), this is reflective of the quality of processing before extraction. Unfortunately, these observations require proof in the form of biosynthetic knowledge, but almost nothing is known about the biosynthesis of oligomeric and polymeric procyanidins in grape seed. Hence, it is necessary to direct studies toward these end goals to take extract optimisation a step forward.

The recognition of the intra- and inter-specific variability of bioactive metabolites has generally been overlooked by both researchers and industry. While the genotypic factors influencing chemical variability are accepted, studies that seek to identify abiotic factors are not conclusive, due to limited control over overlapping variables. Researchers should not be discouraged by this, but greater monitoring of the effects of changing variables should be included in studies, i.e., higher altitude environments may also have different rain regimes. Lastly, given their widespread popularity and high monetary value, the quality, safety, and efficacy of commercial grape seed products represent a fundamental concern. The adulteration of grapeseed dietary supplements is a common problem, particularly with other procyanidin-rich extracts from peanut skin and pine bark. The methods for detecting adulteration prescribed by pharmacopoeias are inadequate to combat this problem. Hence, additional measures are necessary to confirm authenticity and to standardise the chemical profile, particularly in clinical trials when it is necessary to link bioactive ingredients to specific health outcomes.

## Figures and Tables

**Figure 1 plants-11-00809-f001:**
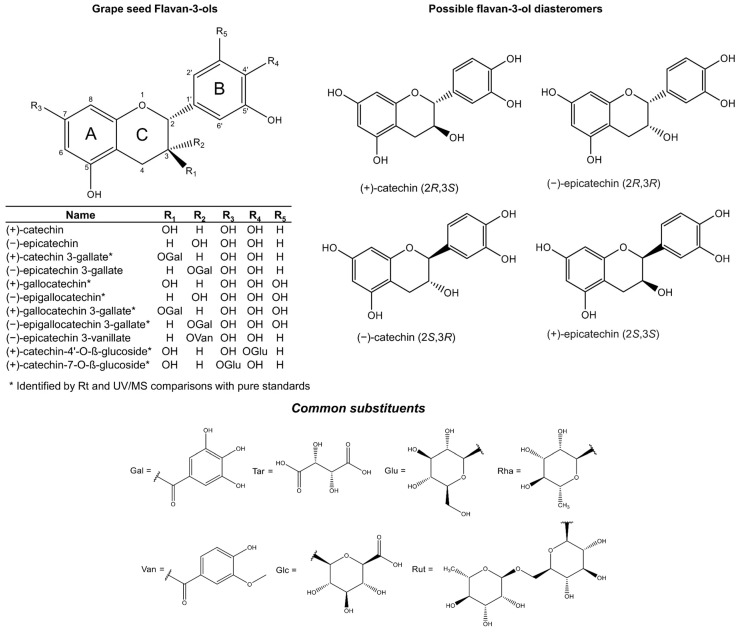
Flavan-3-ol monomers reported in grape seed (upper left) including possible flavan-3-ol epimers and enantiomers (upper right). Common O- or COO-linked substituents of flavan-3-ols and other phenols are included in the lower panel.

**Figure 2 plants-11-00809-f002:**
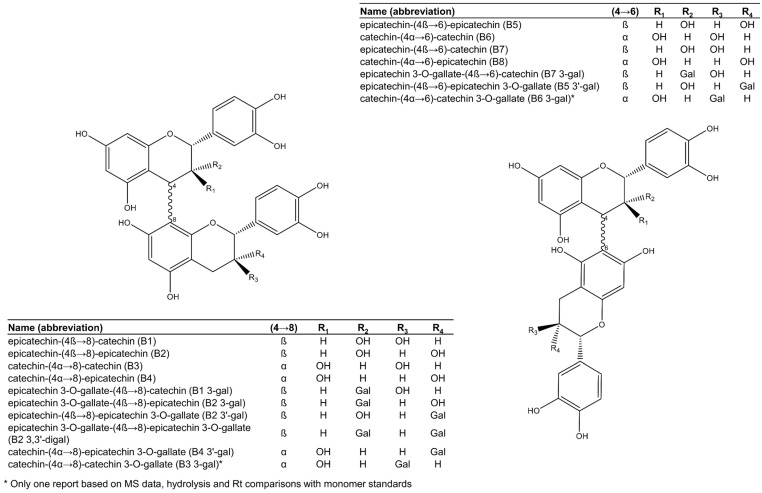
Procyanidin dimers reported in grape seeds.

**Figure 3 plants-11-00809-f003:**
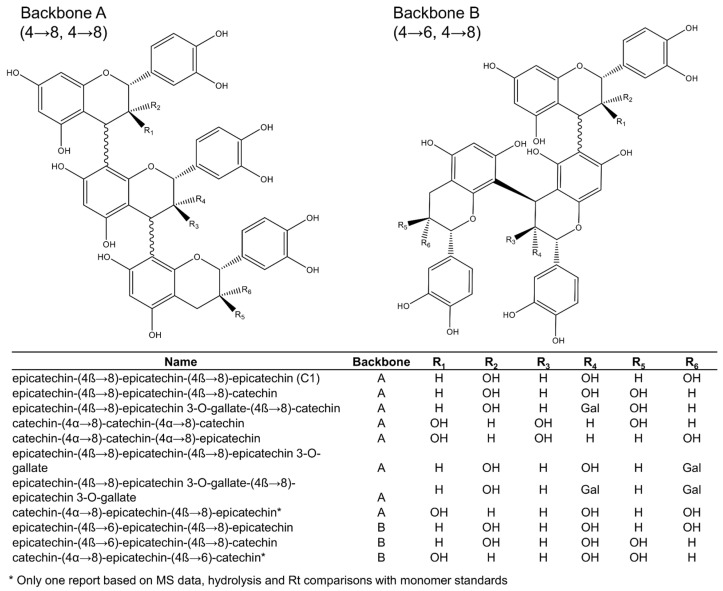
Procyanidin trimers (backbones A and B) reported in grape seeds.

**Figure 4 plants-11-00809-f004:**
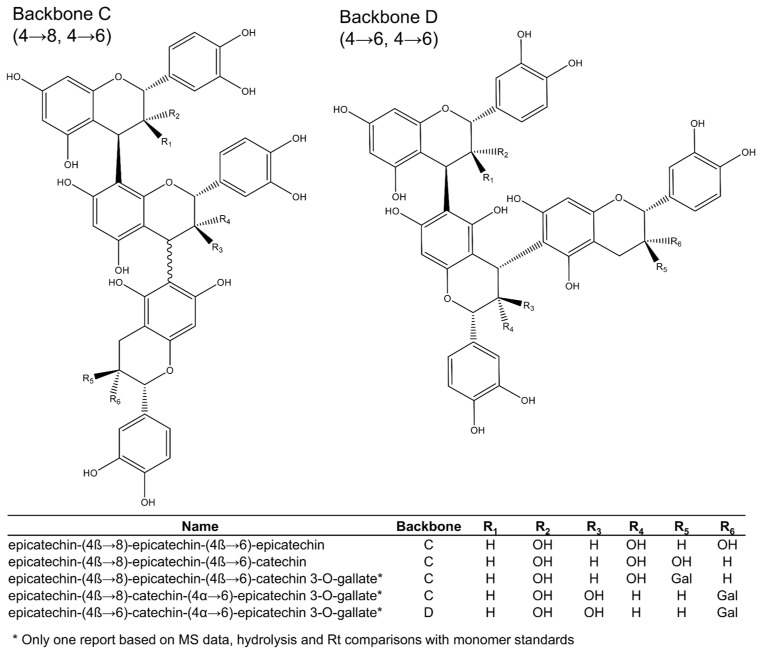
Procyanidin trimers (backbones C and D) reported in grape seeds.

**Figure 5 plants-11-00809-f005:**
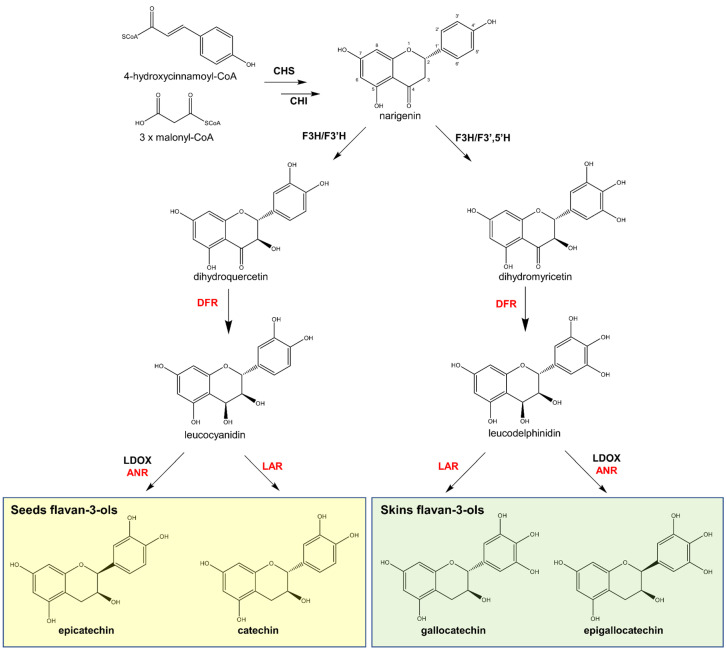
Biosynthetic route of flavan-3-ols in grape. Enzymes: Chalcone synthase (CHS), chalcone isomerase (CHI), flavanone 3-hydroxylase (F3H) and flavanone 3′-hydroxylase (F3′H), flavanone 3′,5′-hydroxylase (F3′5′H), dihydroflavonol 4-reductase (DFR), leucoanthocyanidin dioxygenase (LDOX), and anthocyanidin reductase (ANR) [5,57,58].

**Figure 6 plants-11-00809-f006:**
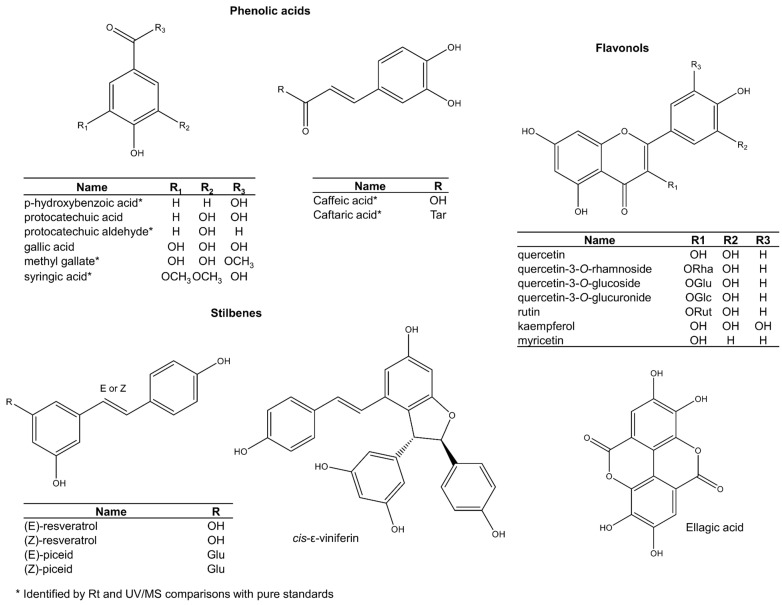
Other phenolic compounds reported in grape seeds.

**Figure 7 plants-11-00809-f007:**
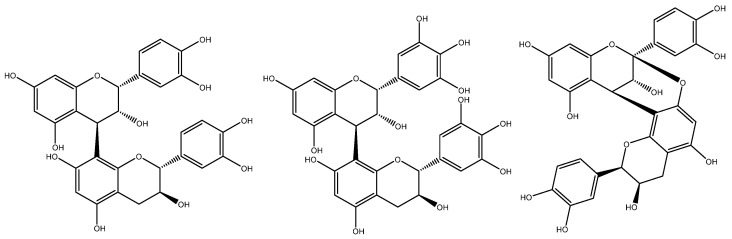
Chemical structures of B-type procyanidins (**left**) characteristic of grape seed, grape skins prodelphinidins (**centre**), and A-type procyanidins (**right**), often found in peanut skin and pine bark.

**Figure 8 plants-11-00809-f008:**
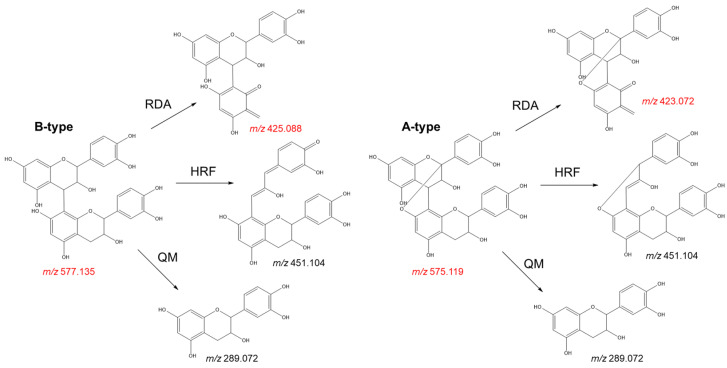
Fragmentation pattern of A-type and B-type procyanidin dimers showing the product ions formed by quinone methide (QM), heterocyclic ring fission (HRF), and Retro-Diels Alder (RDA) fragmentation. Key ions highlighted in red.

## Data Availability

Not applicable.

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
