# Peer review of "Chemical Diversity of Flavan-3-Ols in Grape Seeds: Modulating Factors and Quality Requirements"

_plants, 2022, doi:10.3390/plants11060809_

Round 1
Reviewer 1 Report
This is a very interesting work with potential high practical impact. Some changes are proposed to improve the quality of the text:
-Some writing mistakes can be found in the manuscript – examples in line 14, sentence in lines 38-40, 409, others
-At the end of “Introduction” perhaps authors can explain in higher detail de search strategy (e.g. databases, keywords,…)
-In a non-systematic review, a section of “Results” seem to not be adequate
-It seems that several related reviews were already published and therefore it is important that authors evidence in the manuscript the novelty of this work over others published
-In lines 119-121 authors referred that “They are characterized by the presence of a 2-phenyl-3,4-dihydro-2H-chromene skeleton, which is hydroxylated at position 3 of ring C (Figure 1).”. I think that this is arguable, because in the literature it is possible to find flavanols without a OH in C-ring
-The sentence in lines 363-366 is arguable, because cytochrome enzymes are not phase 2 metabolizers
-As flavonoids, including flavanols, are of high therapeutic interest; however on of the mains limitations of their use is solubility problems and low gastrointestinal absorption. I consider that this point should be developed and I expected to see this in section “2.1.3. Pharmacokinetics and structure-activity relationships”
-Line 410 – “flavonoid” also includes flavanols and therefore this sentence should be changed
-Figure 6 – quercetin, kaempferol and myriceti are not glycosylated – correct figure?
Author Response
Dear Ms. Varittha Sritalahareuthai
Assistant Editor, MDPI Bangkok
Plants
Ref.: plants-1624810
Title: "Chemical Diversity of Flavan-3-Ols in Grape Seeds: Modulating Factors and Quality Requirements"
We would like to thank this editorial and reviewers for their valuable comments and guidance in helping us improve our manuscript. We have closely followed the reviewer’s suggestions and have outlined below point-by-point our responses to the editor’s and reviewers’ comments.
Changes in the original manuscript were made using the “track changes” option according to the instructions to authors. Additional changes in the manuscript were also made to improve clarity.
Answers to Reviewers comments:
Reviewer 1
This is a very interesting work with potential high practical impact. Some changes are proposed to improve the quality of the text:
-Some writing mistakes can be found in the manuscript – examples in line 14, sentence in lines 38-40, 409, others
R/ We would like to thank the reviewer for the kind words and important contribution. We have doubled-checked the text and corrected all the typos and mistakes.
-At the end of “Introduction” perhaps authors can explain in higher detail de search strategy (e.g. databases, keywords,…)
R/ This information was included in lines 114-116.
-In a non-systematic review, a section of “Results” seems to not be adequate
R/ The header “Results” was changed to the more appropriate term “Grape seed chemistry”
-It seems that several related reviews were already published and therefore it is important that authors evidence in the manuscript the novelty of this work over others published
R/ Although several reviews have been published about grape seeds, most of them focus on the biological properties of this nutraceutical and none of them presents a comprehensive review on grape seed chemistry and the factors that modulate it. Thus, the incentive to write the current review was outlined in lines 63-68 “although the biological properties of grape seed polyphenols have been extensively reviewed in recent years, information on the chemical diversity of grape seed flavan-3-ols and the genetic and climatic factors modulating their expression in different grape cultivars is sparce. This information void is problematic because variation in the types of procyanidins present in the seeds can elicit different biological effects”.
-In lines 119-121 authors referred that “They are characterized by the presence of a 2-phenyl-3,4-dihydro-2H-chromene skeleton, which is hydroxylated at position 3 of ring C (Figure 1).”. I think that this is arguable, because in the literature it is possible to find flavanols without a OH in C-ring
R/ If the 2-phenyl-3,4-dihydro-2H-chromene skeleton is not hydroxylated in position C3, then it cannot be characterized as a Flavan-3-ol but a simple “flavan”. Therefore, all flavan-3-ols, by definition and etymology, are hydroxylated in position C3 as observed in all flavanol structures and their polymerization products reported in grape seeds.
-The sentence in lines 363-366 is arguable, because cytochrome enzymes are not phase 2 metabolizers
R/ The sentence was corrected
-As flavonoids, including flavanols, are of high therapeutic interest; however on of the mains limitations of their use is solubility problems and low gastrointestinal absorption. I consider that this point should be developed, and I expected to see this in section “2.1.3. Pharmacokinetics and structure-activity relationships”
R/ This information was included in the relevant section
-Line 410 – “flavonoid” also includes flavanols and therefore this sentence should be changed
R/ The term “flavonol” was introduced as it is more specific in this case.
-Figure 6 – quercetin, kaempferol and myricetin are not glycosylated – correct figure?
R/ The header of the figure was changed.
Reviewer 2 Report
This review deals with the topic of flavanols and their derivatives in grape seeds in a rigorous and well-structured way. In my opinion, this is a good paper that deserves to be published. It is a bit long and tedious in some sections because of its thoroughness, but I do not have critical issues with the paper.
Some general comments that authors could be addressed are as follows:
The article pays a lot of attention to the identification and elucidation of some compounds, both the major and the more specific and minor molecules. This aspect may be important for the characterization and authentication purposes in search of possible adulterations, but I guess that it is so relevant from the point of view of the activity of the nutraceuticals. What is the importance of minor flavanols in the activity of nutraceuticals? Could the authors include some comments on this point in the manuscript?
Regarding authentication, the authors focus exclusively on profiling approaches. Beyond the specific scope of the paper, fingerprinting-based strategies are widely used. In this sense, are there papers based on fingerprinting for grape seed characterization?
Author Response
Reviewer 2
This review deals with the topic of flavanols and their derivatives in grape seeds in a rigorous and well-structured way. In my opinion, this is a good paper that deserves to be published. It is a bit long and tedious in some sections because of its thoroughness, but I do not have critical issues with the paper.
Some general comments that authors could be addressed are as follows:
The article pays a lot of attention to the identification and elucidation of some compounds, both the major and the more specific and minor molecules. This aspect may be important for the characterization and authentication purposes in search of possible adulterations, but I guess that it is so relevant from the point of view of the activity of the nutraceuticals. What is the importance of minor flavanols in the activity of nutraceuticals? Could the authors include some comments on this point in the manuscript?
R/ We would like to thank the reviewer for the time taken to evaluate our manuscript and accurate comments. We decided not to comment on this specific issue because unfortunately there are no reports in the literature comparing the bioactivity of grape seed extracts with and without specific minor flavanols. Most of the studies that link bioactivity to chemistry characterize only the major metabolites. Furthermore, we also consider an extensive interpretation/discussion of this aspect is outside the scope of our review.
Regarding authentication, the authors focus exclusively on profiling approaches. Beyond the specific scope of the paper, fingerprinting-based strategies are widely used. In this sense, are there papers based on fingerprinting for grape seed characterization?
R/ We are not sure what the reviewer means by fingerprinting, as it could be chemical fingerprints or most likely DNA-fingerprinting. If referred to DNA fingerprinting, unfortunately, the grape seed extracts available in the market are produced by extraction methods that usually degrade DNA (i.e., organic solvents or supercritical fluids), and therefore, to the best of our knowledge there are no published papers about this topic.
Reviewer 3 Report
Dear Editor and Authors,
The manuscript ‘Chemical Diversity of Flavan-3-Ols in Grape Seeds: Modulating Factors and Quality Requirements’ by Guillermo F. Padilla-González, Esther Grosskopf, Nicholas J. Sadgrove and Monique S. J. Simmonds is a review on flavan-ols in grapes but the Authors also mention other polyphenols and compounds found in grapes. The manuscript is well written and interesting. Me personally found the chapter about grape polyphenol extract alterations with peanut skin or pine bark, and how to detect those by LC-MS method especially interesting. I recommend publication of the manuscript after a minor revision.
The list of suggested changes:
Line 105 Provide citations on ellagitannin content in grapes (see page 13, you have plenty of them there)
Line 116 ‘Detailed reviews of the chemical constituents in grape seed oil are provided by multiple authors [23-25]’ For me 3 citations is not multiple. Consider deleting the sentence or rewriting , for instance ‘It is worth mentioning that grape seed oil also have many beneficial properties like…’
Line 175 but also before and after. Varieties should be given as ‘Merlot’ ‘Cabernet-Savignon’
Line 528 Citation ‘(Núñez et al., 2006)’ should be given a number, like [56]
Line 656 Change ’Véraison’ into ‘Veraison’
Line 705 Use spaces between number and an unit: 1000 m, and 1500 m
Yours sincerely,
Author Response
Reviewer 3
The manuscript is well written and interesting. Me personally found the chapter about grape polyphenol extract alterations with peanut skin or pine bark, and how to detect those by LC-MS method especially interesting. I recommend publication of the manuscript after a minor revision.
R/ We would like to thank the reviewer for the kind words referring to our manuscript.
The list of suggested changes:
Line 105 Provide citations on ellagitannin content in grapes (see page 13, you have plenty of them there)
R/ A references to the relevant section was included.
Line 116 ‘Detailed reviews of the chemical constituents in grape seed oil are provided by multiple authors [23-25]’ For me 3 citations is not multiple. Consider deleting the sentence or rewriting , for instance ‘It is worth mentioning that grape seed oil also have many beneficial properties like…’
R/ The sentence was edited to avoid ambiguity, and the “multiple authors” part was deleted. Benefits of grape seed oil were not included as it is outside the scope of our review.
Line 175 but also before and after. Varieties should be given as ‘Merlot’ ‘Cabernet-Savignon’
R/ Corrected
Line 528 Citation ‘(Núñez et al., 2006)’ should be given a number, like [56]
R/ Corrected
Line 656 Change ’Véraison’ into ‘Veraison’
R/ Corrected
Line 705 Use spaces between number and a unit: 1000 m, and 1500 m
R/ Corrected